# Effects of Creatine and β-Alanine Co-Supplementation on Exercise Performance and Body Composition: A Systematic Review

**DOI:** 10.3390/nu17132074

**Published:** 2025-06-21

**Authors:** Damoon Ashtary-Larky, Darren G. Candow, Scott C. Forbes, Leila Hajizadeh, Jose Antonio, Katsuhiko Suzuki

**Affiliations:** 1Nutrition and Metabolic Diseases Research Center, Ahvaz Jundishapur University of Medical Sciences, Ahvaz 6135715794, Iran; 2Faculty of Kinesiology and Health Studies, University of Regina, Regina, SK S4S 0A2, Canada; darren.candow@uregina.ca; 3Department of Physical Education Studies, Brandon University, Brandon, MB R7A 6A9, Canada; forbess@brandonu.ca; 4Department of Sport Physiology, Marvdasht Branch, Islamic Azad University, Marvdasht 7371113119, Iran; hajizadehleila266@gmail.com; 5Department of Health and Human Performance, Nova Southeastern University, Davie, FL 33314, USA; jose.antonio@nova.edu; 6Faculty of Sport Sciences, Waseda University, Tokorozawa 359-1192, Japan; katsu.suzu@waseda.jp

**Keywords:** creatine, β-alanine, anaerobic power, maximal strength, endurance, body composition

## Abstract

**Background/Objectives**: Creatine and β-alanine are two widely used dietary supplements known to enhance exercise performance and improve body composition; however, less is known regarding the synergistic effects of combining the two supplements. **Methods**: A systematic search was conducted across PubMed/MEDLINE, Scopus, and Web of Science databases for randomized controlled trials (RCTs) published up to March 2025. Eligible studies included adult participants receiving creatine and β-alanine together compared to creatine or β-alanine alone for at least four weeks and assessed measures of exercise performance and/or body composition. Study quality was assessed using the Cochrane Risk of Bias tool. **Results**: A total of 7 randomized controlled trials (*n* = 263 participants; 231 males and 32 females) met the inclusion criteria. Collectively, the combination of creatine and β-alanine supplementation enhanced high-intensity exercise performance, particularly anaerobic power and repeated-bout performance, compared to creatine or β-alanine alone. Co-ingestion of creatine and β-alanine supplementation did not increase measures of maximal strength compared to creatine alone. The effects of creatine and β-alanine supplementation on body composition were equivocal, with one study reporting greater lean mass gains and fat mass reductions compared to creatine and β-alanine supplementation individually, while another found no significant improvements. Additionally, no significant improvements in aerobic endurance capacity (VO_2_max, lactate threshold, or time to exhaustion) were observed from creatine and β-alanine supplementation co-ingestion. **Conclusions**: The combination of creatine and β-alanine supplementation may be effective for enhancing high-intensity exercise performance but has no greater effect on maximal strength, body composition, or measures of aerobic capacity compared to creatine or β-alanine alone.

## 1. Introduction

Creatine and β-alanine are among the most widely utilized ergogenic supplements to enhance exercise performance [1,2]. Creatine supplementation is well-established for its role in increasing intramuscular phosphocreatine stores, thereby enhancing rapid adenosine triphosphate (ATP) resynthesis during high-intensity, short-duration exercise [3]. Further, creatine has been shown to enhance protein kinetics, satellite cells, and myogenic regulatory factors and attenuate myostatin, inflammation, and oxidative stress [4]. These mechanisms contribute to improvements in power output, and chronic creatine supplementation—typically involving a loading phase of approximately 20 g/day for 5–7 days, followed by a maintenance dose of 2–5 g/day—has been associated with greater gains in lean body mass (LBM) and strength during resistance training [5].

β-Alanine, a non-essential amino acid, serves as a precursor to carnosine [6,7], a key intramuscular buffer that helps regulate exercise-induced acidosis [8]. Chronic β-alanine supplementation (typically 3.2–6.4 g/day for 2–10 weeks) increases muscle carnosine content by approximately 10% after 2 weeks and up to 80% following 10 weeks of supplementation [9]. This elevation in intramuscular carnosine enhances the muscle’s buffering capacity, potentially delaying the onset of fatigue and improving high-intensity performance lasting between 1 and 4 min [10].

Given their physiological mechanisms, co-supplementation with creatine and β-alanine has been purported to elicit complementary or potentially synergistic ergogenic effects. Preliminary investigations into the combined effects of creatine and β-alanine supplementation have reported promising but mixed results. Zoeller et al. suggested that co-supplementation with creatine and β-alanine may improve ventilatory threshold (VT) and lactate threshold (LT) during incremental cycling exercise in male participants [11]. Additionally, in the context of strength and power performance, Hoffman et al. reported that 10 weeks of creatine and β-alanine supplementation during resistance training conferred “independent and potentially synergistic” benefits in male athletes [12]. However, despite these positive findings, the evidence remains inconclusive, as some studies have failed to observe additive effects of combining these supplements [9]. Given the variability in outcomes, a systematic review of randomized controlled trials (RCTs) is warranted to elucidate the combined effects of creatine and β-alanine supplementation on exercise performance and body composition. Therefore, this review aims to critically assess the evidence from RCTs on co-supplementation with creatine and β-alanine, with a specific focus on key performance outcomes (strength, power, muscular endurance, and aerobic capacity) and body composition changes (lean mass and fat mass). Given the variability in study designs and methodologies, the findings are synthesized qualitatively, without conducting a meta-analysis, to provide a comprehensive and up-to-date evaluation of the potential additive benefits of combined creatine and β-alanine supplementation. This review seeks to determine whether co-supplementation offers superior ergogenic effects compared to either supplement alone, thereby providing evidence-based insights for athletes, coaches, and sports practitioners regarding its efficacy and the certainty of available evidence. Based on previous studies, we hypothesized that co-supplementation with creatine and β-alanine would have beneficial effects on exercise performance and body composition compared to either supplement alone.

## 2. Materials and Methods

This systematic review was conducted in accordance with the Preferred Reporting Items for Systematic Reviews and Meta-Analyses (PRISMA) guidelines, ensuring transparent and rigorous reporting [13]. A predefined protocol was established to outline the objectives, inclusion criteria, and methodology for evidence synthesis. This systematic review protocol was registered with the International Prospective Register of Systematic Reviews (PROSPERO) under the registration number CRD420251066302.

### 2.1. Eligibility Criteria

Studies were included if: (1) population: adults (athletes or untrained) of any sex; (2) intervention: combined supplementation with creatine and β-alanine, administered concurrently for at least 4 weeks (or a duration sufficient to load both supplements), alone or in addition to a training program; multi-ingredient interventions were excluded unless the co-interventions were matched between groups and the only difference was the presence or absence of creatine and β-alanine, allowing isolation of its effects; (3) comparators: placebo or single-supplement groups (creatine-only or β-alanine-only) in a randomized design; (4) outcomes: exercise performance measures (strength, power, endurance, aerobic capacity, etc.) and/or body composition (LBM, fat mass, or body fat percentage); (5) study design: randomized controlled trials only.

Studies were excluded if they investigated acute, single-dose supplementation rather than chronic supplementation; utilized non-randomized designs, observational methodologies, or case studies; or examined multi-ingredient supplements where additional ergogenic substances (e.g., citrulline) were co-administered—unless those ingredients were matched across control groups to isolate the effects of creatine and β-alanine. No restrictions were placed on participant training status or the specific training protocols used, provided that all groups within a study followed comparable training conditions.

### 2.2. Information Sources and Search Strategy

A comprehensive literature search was performed using PubMed/MEDLINE, Scopus, and Web of Science from inception through March 2025. The following keywords and Boolean combinations were used: “creatine AND beta-alanine”, “β-alanine AND creatine AND performance”, “combined supplementation”, “carnosine”, “supplementation”, “co-supplementation”, “strength”, “one-repetition maximum”, “1RM”, “endurance”, “body composition”, “VO_2_max”, “weight loss”, “lean mass”, and “randomized trial”. Reference lists of relevant papers, including previous reviews and position stands, were hand-searched for any additional RCTs. No language restrictions were applied, although all identified studies were in English. Database search results were merged, and duplicates were removed using reference management software (EndNote X9 software).

### 2.3. Study Selection

The titles and abstracts of retrieved studies were screened to determine their relevance. Full-text articles deemed potentially eligible were then assessed against the predefined inclusion criteria. In cases where eligibility was uncertain, careful reconsideration of the inclusion and exclusion criteria was conducted to ensure objective selection.

### 2.4. Data Extraction

For each included trial, descriptive data were extracted, including sample size, participant characteristics, training status, supplementation protocol (dosing and duration), training program (if applicable), and comparator groups. Outcome measures related to exercise performance (e.g., one-repetition maximum (1RM) strength, Wingate anaerobic power, jump performance, time-to-exhaustion (TTE), and VO_2_max) and body composition (e.g., body mass, lean mass, fat mass, and body fat percentage) were recorded. When available, the magnitude and statistical significance of changes in these outcomes were extracted, particularly in the co-supplementation group compared to control or single-supplement groups.

### 2.5. Risk of Bias Assessment

The risk of bias was evaluated using the Cochrane risk-of-bias assessment tool, which considers multiple methodological domains, including randomization process, allocation concealment, participant and staff blinding, outcome assessor blinding, completeness of outcome data, selective reporting, and other sources of bias. Based on these criteria, studies were categorized into three groups: high risk of bias (≥2 domains rated as high risk), unclear risk of bias (exactly two domains rated as high risk), and low risk of bias (<2 domains rated as high risk).

### 2.6. Synthesis of Results

Due to the heterogeneity in study designs and outcome measures, a meta-analysis was not conducted. Instead, a qualitative synthesis was performed, organizing findings into distinct outcome domains: strength and power performance, endurance performance, and body composition. Results from individual studies were systematically reported and compared, emphasizing whether co-supplementation with creatine and β-alanine yielded superior improvements relative to placebo or single-supplement conditions. Where relevant, effect directions and statistical significance were documented. The narrative synthesis integrates these findings with the quality of evidence to derive overall conclusions regarding the efficacy of combined supplementation.

## 3. Results

### 3.1. Study Characteristics

As illustrated in Figure 1, a total of seven RCTs published between 2003 and 2022 met the inclusion criteria, collectively enrolling 263 participants [9,11,12,14,15,16,17]. Study characteristics are summarized in Table 1. Participants were between 20 and 30 years of age. Six studies included male participants only [11,12,14,15,16,17], while one study exclusively examined female participants [9]. The supplementation protocols adhered to standard dosing strategies for creatine and β-alanine. For creatine, most studies implemented a loading phase (~20 g/day or 0.3 g/kg/day, divided into multiple doses over 5–7 days), followed by a maintenance dose (~5–10 g/day) or a body mass-adjusted dose (e.g., 0.1 g/kg/day) [9,11,15,17]. However, some studies employed only a loading phase [14] or a maintenance protocol without an initial loading phase [12], or, in one case, a reversed approach where a maintenance dose was administered before the loading phase [16]. β-Alanine dosing ranged from 3.2 g/day to 6.4 g/day [9,11,12,14,15,16,17], typically divided into multiple daily doses to enhance tolerance and minimize paresthesia. The duration of combined supplementation varied between 4 and 10 weeks.

All studies included at least one single-supplement comparison group (creatine-only and/or β-alanine-only) to assess the potential additive effects of co-supplementation. Additionally, all trials except one study [14] incorporated a placebo control group (e.g., dextrose) alongside at least one combined creatine plus β-alanine group.

### 3.2. Results of Quality Assessment

The quality assessment of included studies, presented in Figure 2, reveals variability in the risk of bias across different methodological aspects. While most studies demonstrated a low risk of bias in key domains such as participant blinding, outcome assessor blinding, and allocation concealment, some exhibited unclear or high risk in specific categories.

Notably, Harris et al. [17], Zoeller et al. [11], and Okudan et al. [16] had an unclear risk of bias in random sequence generation, indicating potential limitations in randomization procedures. Additionally, selective reporting and incomplete outcome data emerged as the most frequent sources of high risk of bias, particularly in Stout et al. [15] and Samadi et al. [14].

Overall, based on the Cochrane scoring system, studies with more than two high-risk domains were classified as high risk of bias, while those with exactly two high-risk factors were categorized under unclear risk. Studies exhibiting less than two high-risk domains were considered low risk of bias. These findings emphasize the need for greater methodological rigor in future research to enhance the reliability and validity of reported outcomes.

### 3.3. Outcomes Measured

The primary outcomes assessed across studies included maximal strength (1RM in bench press and squat), anaerobic power (Wingate test outputs, vertical jump), high-intensity exercise capacity (repeated sprint performance, TTE at a fixed intensity), aerobic capacity (VO_2_max or VO_2_peak), and body composition (fat mass and lean mass). Findings are reported sequentially, beginning with strength and power performance, followed by endurance/aerobic performance, and concluding with body composition outcomes.

#### 3.3.1. Strength and Power Performance

Five trials evaluated muscular strength adaptations (primarily 1RM in resistance exercises) and anaerobic power output following co-supplementation with creatine and β-alanine [9,12,14,16,17]. Findings related to maximal strength are summarized first, followed by studies examining anaerobic power and high-intensity performance.

##### Maximal Strength

The longest-duration study investigating the effects of creatine and β-alanine co-supplementation was conducted by Hoffman et al., who examined collegiate male football players undergoing 10 weeks of heavy resistance training while receiving either a placebo, creatine alone, or creatine + β-alanine [12]. As expected, the creatine-supplemented group exhibited significantly greater gains in 1RM strength (bench press and squat) compared to the placebo group over the training period. Notably, the combined creatine + β-alanine group demonstrated similar strength improvements to the creatine-only group, with both supplement conditions significantly outperforming the placebo in 1RM gains. However, there was no statistically significant difference in strength gains between the combination group and creatine-only group, suggesting that β-alanine did not provide additional benefits for maximal strength beyond the established ergogenic effects of creatine. These findings indicate that, in the context of resistance-trained male athletes, creatine is the primary contributor to maximal strength improvements, while β-alanine does not further enhance 1RM performance when training and diet are controlled.

A more recent study by Samadi et al. investigated β-alanine as the primary intervention, followed by a short-term creatine loading phase. In this 4-week trial, male military personnel initially supplemented β-alanine (6.4 g/day) for three weeks before being randomized to either continue β-alanine alone or add high-dose creatine (~0.3 g/kg/day) during the final week [14]. The creatine + β-alanine group exhibited significant improvements in vertical jump performance, a key indicator of lower-body power, whereas no such improvement was observed in the β-alanine-only group. The authors also reported trends toward greater muscular strength gains in the creatine + β-alanine group, aligning with previous findings that creatine is a potent enhancer of strength and power. However, the addition of creatine did not result in significant improvements in peak power, average power, minimum power, leg press, or chest press compared to β-alanine alone. These results collectively suggest that while co-supplementation enhances maximal strength compared to no supplementation, this effect is primarily attributable to creatine, with β-alanine providing no additional advantage for 1RM strength gains beyond those achieved with creatine and resistance training alone.

##### Anaerobic Power and High-Intensity Performance

Harris et al. conducted the first study examining the co-ingestion of creatine and β-alanine [17]. Their findings revealed that power output during a 4-min cycling test was greater in the creatine + β-alanine group than in the creatine-only group, demonstrating that β-alanine supplementation enhanced high-intensity exercise performance beyond creatine alone. Specifically, participants receiving β-alanine exhibited a significant increase in maximal power output during the 4-min all-out ergometer test, with most of the performance gains occurring within the first minute of exercise, prior to full cardiovascular adjustment. This improvement is likely attributed to an increased hydrogen ion (H^+^) buffering capacity, facilitated by elevated muscle carnosine levels. However, β-alanine supplementation did not significantly affect isometric endurance or overall force output compared to control groups. These findings suggest that co-supplementation with creatine and β-alanine primarily enhances short-duration, high-intensity performance, whereas its effects on endurance-related strength parameters remain limited compared to creatine alone.

In contrast, Hoffman et al. assessed anaerobic power using a 30-s Wingate test and a 20-repetition jump test in their 10-week study. Neither the creatine group nor the creatine + β-alanine group demonstrated significant improvements in Wingate anaerobic power or the 20-repetition jump test [12]. However, both supplement groups exhibited significant gains in 1RM squat strength, 1RM bench press strength, and average weekly training intensity (% 1RM squat) compared to the placebo, with no significant differences between the creatine-only and creatine + β-alanine groups. The only exception was average weekly training intensity, where the co-supplementation group improved significantly compared to the placebo, suggesting a potential benefit of β-alanine in sustaining training intensity over time.

Further evidence comes from Okudan et al., who conducted a 28-day RCT in untrained young men, specifically investigating repeated Wingate sprint performance across four groups: placebo, creatine-only, β-alanine-only, and combined β-alanine + creatine [16]. Participants completed three consecutive 30-s Wingate tests (separated by brief rest periods) before and after the supplementation period to simulate repeated supramaximal efforts. As expected, the placebo group experienced performance decrements across sprints in both pre- and post-supplementation tests, indicating fatigue accumulation. In contrast, the creatine-only group exhibited significant improvements in peak power output during the second and third sprints, suggesting better maintenance of anaerobic power over repeated efforts. However, the β-alanine + creatine group showed an additional advantage, as they significantly increased mean power output across all three sprints—an improvement not observed in placebo or β-alanine-only conditions. Furthermore, the fatigue index (rate of power decline across sprints) remained stable in groups receiving β-alanine, indicating reduced fatigue accumulation. These findings suggest that while creatine improves peak power, β-alanine contributes to sustained power output and mitigates fatigue in repeated high-intensity efforts.

A study by Kresta et al. in recreationally active women further examined repeated-sprint performance following a 4-week supplementation period with β-alanine, creatine, a combination of both, or a placebo [9]. Participants performed multiple Wingate anaerobic capacity tests before, during, and after supplementation. Unlike previous findings in men, the co-supplementation group did not exhibit significant advantages over the single-supplement groups in any performance measure. The authors concluded that co-supplementation did not provide additive benefits for anaerobic performance in female participants. Additionally, all groups, including the placebo group, experienced some performance improvements with training, further reinforcing the lack of significant differences between supplementation conditions.

#### 3.3.2. Endurance and Aerobic Performance

In this review, the term “endurance” refers to aerobic or high-intensity exercise capacity, including VO_2_max, VT, LT, and TTE during sustained exercise. The potential synergistic effects of co-supplementation with creatine and β-alanine on aerobic capacity and endurance performance have been examined in several studies, which are presented below.

##### Aerobic Power (VO_2_max/VO_2_peak)

The impact of creatine and β-alanine co-supplementation on maximal aerobic capacity (VO_2_max or VO_2_peak) has been explored in a limited number of trials. Zoeller et al. conducted an RCT with 55 male participants, examining 4 weeks of supplementation with placebo, creatine, β-alanine, or a combination of both [11]. The study assessed VO_2_peak using a graded exercise test on a cycle ergometer, along with VT, LT, and TTE at VO_2_peak. The results indicated no significant differences between groups for any of the primary endurance-related outcomes—neither creatine, β-alanine, nor their combination produced statistically significant improvements in VO_2_peak or TTE compared to the placebo over 4 weeks. However, the co-supplementation group exhibited the most consistent within-group improvements, showing significant pre-post increases in five of the eight endurance parameters measured (e.g., a trend toward increased VO_2_peak and delayed fatigue onset), whereas the single-supplement groups demonstrated improvements in fewer parameters (one in the β-alanine group and two in the creatine group). As the placebo group also showed minor improvements—likely due to training adaptation or test familiarization—the between-group differences did not reach statistical significance. The authors cautiously suggested that co-supplementation “may potentially enhance endurance performance”, though the evidence for a true additive effect was weak. From a practical standpoint, any enhancement in VO_2_max by these supplements appears to be minimal.

Similarly, Kresta et al., in a study of female participants, found no significant benefit of creatine + β-alanine co-supplementation on VO_2_peak [9]. Following 4 weeks of supplementation combined with endurance and cycle training, changes in VO_2_peak did not significantly differ between the β-alanine, creatine, combined, or placebo groups. Notably, β-alanine supplementation alone did not outperform the placebo for aerobic capacity, aligning with previous research in male subjects that also failed to demonstrate a significant impact of β-alanine on VO_2_max. Although creatine alone showed a non-significant trend toward improved TTE at VO_2_peak in the female cohort, the effect was minor. Overall, co-supplementation failed to improve maximal aerobic fitness beyond the effects of training alone, with the authors concluding that their findings support prior literature indicating that creatine has, at best, a modest and inconsistent impact on aerobic capacity.

##### Submaximal Endurance and Fatigue Thresholds

Several studies have explored whether co-supplementation with creatine and β-alanine influences the onset of fatigue during incremental exercise. As previously noted, Zoeller et al. assessed ventilatory and LT and found no significant group differences, although the combined supplementation group exhibited the largest numerical improvements [11].

A separate study by Stout et al. focused on neuromuscular fatigue thresholds, using a metric known as physical working capacity at the fatigue threshold (PWCFT), derived from electromyographic signals during incremental cycling [15]. In this 28-day trial in untrained men, participants were assigned to one of four groups: placebo, creatine, β-alanine, or creatine + β-alanine. The β-alanine-only and co-supplementation groups both exhibited significant improvements in fatigue threshold compared to the placebo, whereas creatine alone did not. Importantly, the increase in fatigue threshold in the co-supplementation group was not significantly greater than in the β-alanine-only group, indicating that adding creatine did not provide additional benefits for neuromuscular fatigue resistance beyond those conferred by β-alanine alone. The authors concluded that β-alanine was effective in delaying neuromuscular fatigue, while creatine did not contribute meaningfully to this endurance-related outcome. These findings reinforce the notion that β-alanine’s buffering capacity plays a crucial role in sustaining high-intensity effort, whereas creatine’s role in endurance-type performance is minimal.

##### Time-to-Exhaustion and Thresholds

Several studies have examined the effects of co-supplementation on TTE at submaximal intensities and related endurance measures. As mentioned, Zoeller et al. found no significant group differences in TTE, though the co-supplementation group exhibited the largest numerical increase [11]. Similarly, Kresta et al. found no significant improvements in TTE or LT across groups [9].

In Kresta’s female cohort, β-alanine supplementation for 4 weeks did not significantly increase the LT, though a trend toward improved lactate dynamics was observed after one week in the β-alanine groups. Additionally, Kresta et al. reported a notable finding regarding peak blood lactate levels during the VO_2_max test—the β-alanine-only group exhibited significantly higher peak blood lactate than the co-supplementation and placebo groups post-supplementation. This observation suggests a potential enhancement in anaerobic glycolytic capacity in the β-alanine-alone group, though it remains unclear why the co-supplementation group did not exhibit the same response. The lack of significant differences in VT across groups further supports the conclusion that neither creatine nor β-alanine meaningfully enhances endurance-related parameters in the context of aerobic capacity.

#### 3.3.3. Body Composition Outcomes

Data on the effects of co-supplementation with creatine and β-alanine on body composition are limited, as only two studies have investigated this topic. The findings from these two RCTs are presented below.

##### Lean Body Mass Gains

The most notable finding regarding LBM enhancement comes from the 10-week RCT by Hoffman et al., which examined resistance-trained men undergoing supplementation with creatine, β-alanine, a combination of both, or the placebo [12]. Body composition was assessed using dual-energy X-ray absorptiometry (DEXA). The results demonstrated that the combined creatine + β-alanine group experienced significantly greater gains in LBM compared to both the creatine-only and placebo groups. On average, the co-supplementation group accumulated more LBM than the creatine-alone group, suggesting a potential synergistic effect on muscle hypertrophy or lean tissue retention. Furthermore, the combined group also exhibited a greater reduction in body fat percentage, indicating a possible repartitioning effect, where increases in muscle mass were accompanied by a reduction in fat mass. The authors acknowledged that creatine alone was effective in increasing LBM relative to the placebo, as expected, but the addition of β-alanine appeared to further augment lean mass accrual.

Conversely, a similarly designed study in female participants failed to replicate these findings. Kresta et al. assessed body composition using DXA in college-aged women following 4 weeks of supplementation combined with concurrent training [9]. Their results revealed no significant differences between the creatine + β-alanine group and any other group in terms of changes in LBM or fat mass. All groups, including the placebo, exhibited modest improvements in body composition, likely due to the effects of training adaptation (i.e., slight increases in lean mass and decreases in fat mass). However, none of the supplementation conditions significantly enhanced body composition beyond the effects of training alone in women.

##### Fat Mass, Body Fat Percentage, and Body Mass

Changes in fat mass were less frequently reported across studies, though they generally followed an inverse relationship to lean mass changes. In Hoffman et al.’s study, the creatine + β-alanine group not only exhibited greater increases in LBM but also demonstrated a significantly greater reduction in body fat percentage compared to the creatine-only and placebo groups [12]. Over 10 weeks of resistance training, the combined supplementation group maintained stable body mass yet experienced a more pronounced decline in absolute fat mass and body fat percentage. This effect may be attributed to the greater increase in lean mass, which lowered the percentage of body fat, or alternatively, an increase in training volume and energy expenditure due to the ergogenic benefits of supplementation. By contrast, Kresta et al. reported no significant differences in fat mass, body fat percentage, or total body mass between groups in their 4-week trial in female participants [9].

## 4. Discussion

This systematic review aimed to evaluate whether co-supplementation with creatine and β-alanine provides greater improvements in exercise performance and body composition compared to either supplement alone. The findings from RCTs suggest that co-supplementation may be an effective strategy for enhancing certain performance outcomes, particularly strength and high-intensity power output. However, these benefits appear to be primarily driven by the well-established ergogenic effects of creatine. True synergistic effects—where the combination of creatine and β-alanine leads to performance enhancements beyond what either supplement achieves alone—were observed more frequently in anaerobic power and high-intensity performance, with a lower probability of additive effects in maximal strength and body composition. Current evidence regarding the effects of creatine and β-alanine co-supplementation on body composition is both limited and inconsistent. While one study reported greater improvements in lean mass and fat mass reduction compared to individual supplementation, another found no significant changes. Furthermore, co-supplementation did not result in meaningful improvements in aerobic endurance measures, including VO_2_max, lactate threshold, or TTE.

### 4.1. Anaerobic Performance

It is well established that creatine supplementation enhances strength and power [18,19], and all RCTs reviewed in this study confirmed that groups receiving creatine (either alone or in combination) exhibited substantial improvements in strength and anaerobic power relative to the placebo [12,14]. These findings are consistent with decades of research demonstrating that creatine is one of the most effective supplements for increasing muscular strength, primarily by enhancing training capacity, phosphocreatine resynthesis, and neuromuscular function [20,21,22,23].

One of the primary mechanisms by which β-alanine is hypothesized to enhance strength and power performance is through its buffering effect on intramuscular acidity [24,25]. The accumulation of H^+^ during high-intensity exercise has been shown to impair phosphocreatine resynthesis [26], inhibit glycolytic metabolism [27], and negatively affect muscle contractile function. Thus, it is theoretically plausible that combining creatine with β-alanine could provide additional benefits by simultaneously supporting ATP production (creatine) and buffering metabolic acidosis (β-alanine). However, the question remains whether this combination meaningfully enhances strength and power outcomes beyond what creatine alone achieves.

The evidence from RCTs suggests that any additional benefit of β-alanine in maximal strength gains is minimal. Hoffman et al. found no significant differences in 1RM strength gains (bench press and squat) between the creatine-only and creatine + β-alanine groups, indicating that β-alanine did not further enhance maximal strength beyond the effects of creatine [12]. Other studies that assessed strength outcomes similarly reported that creatine alone was superior to the placebo but did not specifically compare creatine-only vs. creatine + β-alanine for 1RM gains. These results suggest that creatine’s mechanism (increasing phosphocreatine availability) is already sufficient to maximize strength adaptations in a resistance training context and that additional buffering from β-alanine does not provide a substantial advantage for maximal strength performance.

In contrast, β-alanine may offer greater benefits in scenarios where repeated maximal efforts are required, such as high-intensity interval training (HIIT), sprinting, or repeated sets of resistance exercise. This is evident in studies where muscle acidosis is a key limiting factor. For example, Okudan et al. examined repeated Wingate sprint performance in untrained men and found that while creatine improved peak power, only the groups receiving β-alanine (β-alanine alone or β-alanine + creatine) were able to sustain power output across multiple sprints and resist fatigue [16]. This suggests that β-alanine’s buffering effect translated into reduced performance decrements over successive high-intensity efforts, whereas creatine alone was primarily effective for increasing peak power output. The physiological basis for this synergy is logical: Creatine supplementation enhances ATP-PCr system efficiency, providing energy for short-duration, high-intensity efforts. β-Alanine, via its role in carnosine synthesis, buffers H^+^ accumulation, mitigating the decline in pH associated with anaerobic glycolysis and thereby delaying fatigue in subsequent efforts. This complementary interaction likely explains why β-alanine’s effects are more apparent in repeated-bout performance rather than in single maximal lifts (e.g., 1RM strength testing).

A recent study by Li et al. in competitive male basketball players provides further insights into the potential benefits of co-supplementation for explosive performance. This 4-week RCT included the placebo, creatine-only, β-alanine-only, L-citrulline-only, or a combination of all three supplements, alongside a sprint interval training program [28]. The study reported that all supplement groups outperformed the placebo in physical performance tests (e.g., vertical jump, 20-m sprint, and change-of-direction speed) and that the combination of multiple supplements provided the broadest performance enhancements. However, because the co-supplementation group in this study included L-citrulline, these results were not included in the current systematic review due to the predefined inclusion criteria.

Despite this limitation, the findings reinforce that creatine and β-alanine—either alone or in combination—are beneficial for power-related outcomes. Additionally, no negative interactions were observed between multiple supplements, suggesting that co-supplementation strategies may provide broad ergogenic benefits without interference.

It is important to note that not all studies demonstrated a clear differentiation in the roles of creatine and β-alanine. For instance, in Kresta et al.’s study on female participants, even repeated Wingate tests did not reveal a distinct advantage of co-supplementation over single-supplement conditions [9]. Several factors may contribute to these findings. One potential explanation is the influence of training status and sex-related differences. The female participants in Kresta’s study, despite being recreationally active, likely had lower baseline power outputs and muscle mass compared to male athletes. Consequently, the absolute physiological stress and accumulation of metabolic byproducts (e.g., hydrogen ions) during repeated sprints may have been lower, reducing the potential ergogenic effects of β-alanine. Additionally, the dosing strategy (relative to body weight) and the supplementation duration (4 weeks) may have been insufficient for smaller female participants to achieve a meaningful increase in muscle carnosine levels, thereby limiting the impact of β-alanine on performance.

Another relevant consideration is that creatine’s effects on lean mass and power output in women are often less pronounced than in men, potentially due to hormonal differences, muscle fiber composition, or lower total muscle creatine stores [9]. Moreover, Kresta et al. incorporated concurrent endurance training across all study groups, which may have attenuated strength and power adaptations and masked potential supplement effects. This contrasts with Okudan et al.’s study in untrained men, which specifically targeted anaerobic performance (Wingate sprint tests) without endurance training [16]. The differences in training modality and population characteristics suggest that the presence of synergy between creatine and β-alanine may be more evident in anaerobic-focused interventions (e.g., Okudan et al.) but less pronounced in endurance-based or mixed-training contexts (e.g., Kresta et al.).

Another possible reason for inconsistent findings is the presence of ceiling effects [29]. In highly trained populations, such as Hoffman et al.’s collegiate football players, substantial improvements in strength and power are already achieved through training and creatine supplementation alone. In such cases, β-alanine’s ability to further enhance performance may be limited, particularly over a relatively short intervention period (e.g., 10 weeks), where creatine and structured training have already maximized neuromuscular adaptations. Conversely, in less-trained individuals, the potential for improvement may be greater, and a supplement that allows for even marginal increases in training capacity (e.g., an additional repetition or a slight delay in fatigue) could lead to measurable performance differences. This may explain why β-alanine’s effects were more evident in studies involving untrained participants engaging in repeated high-intensity efforts but less pronounced in well-trained athletes undergoing structured resistance training.

The cumulative evidence from RCTs indicates that co-supplementation with creatine and β-alanine enhances strength and power performance relative to the placebo, particularly when combined with structured training programs. However, these improvements are primarily attributed to creatine’s well-established effects on muscular strength and power output. In settings where creatine alone already optimizes neuromuscular performance, β-alanine does not appear to provide additional benefits for maximal strength (e.g., 1RM gains). However, β-alanine may offer advantages in high-intensity, repeated-effort scenarios, where it can delay fatigue and sustain power output over successive efforts.

From a practical standpoint, athletes engaged in strength and power training can expect substantial performance enhancements from creatine alone. Adding β-alanine may be beneficial in scenarios requiring sustained power output over multiple sets or repeated bouts of high-intensity exercise, but its contribution to absolute strength gains appears to be minimal. Future research should explore the long-term effects of co-supplementation across different training populations, with a particular focus on sex-based differences, training status, and sport-specific demands.

### 4.2. Aerobic Performance

The findings of this review provide limited evidence supporting the efficacy of co-supplementation with creatine and β-alanine in enhancing traditional endurance performance metrics, such as VO_2_max or prolonged aerobic exercise capacity, beyond the effects of training alone. This outcome is expected, as creatine supplementation does not inherently improve oxidative metabolism and may even be detrimental to endurance performance if it leads to rapid weight gain without direct benefits to aerobic energy production [30,31]. Conversely, β-alanine’s primary ergogenic role is observed in high-intensity exercise lasting approximately 1–4 min, such as an 800–1500 m run or a 1000 m rowing effort, rather than in prolonged endurance events such as marathons [32]. The current body of evidence confirms that VO_2_max remains unaffected by β-alanine or creatine supplementation [9,11]. However, for efforts occurring at the metabolic “crossover” between anaerobic and aerobic energy systems, there may be a modest indication of benefit. Stout et al. reported that β-alanine supplementation increased the PWCFT, suggesting an improvement in fatigue resistance [15]. However, the addition of creatine did not further amplify this effect, indicating that β-alanine alone was responsible for the observed improvements. Similarly, Zoeller et al. reported modest improvements in TTE at high intensity in the combined supplementation group, though these improvements were not significantly greater than those observed with β-alanine alone [11]. These findings suggest that β-alanine may provide a slight advantage in events involving sustained high-intensity efforts, such as 2–3-min maximal exertion bouts or sports requiring repeated bursts of high-intensity activity (e.g., boxing rounds, middle-distance track races, or 100–200 m swimming sprints). However, direct evidence from studies specifically targeting such performance scenarios remains lacking.

The findings from Li et al. in competitive basketball players further support the limited impact of co-supplementation on aerobic endurance [28]. In this 4-week study, athletes received creatine, β-alanine, a combination of both, or an additional L-citrulline supplementation regimen alongside their training program. Aerobic fitness was assessed (likely via a shuttle run or VO_2_max test), and while all groups improved due to training, the supplemented groups (including the combined group) outperformed the placebo group in general. However, the co-supplementation group did not demonstrate a clear advantage over the single-supplement groups for aerobic performance, suggesting that creatine and β-alanine do not synergistically enhance aerobic endurance when combined.

Another intriguing finding comes from Samadi et al., who observed improved cognitive performance (measured via mathematical processing tasks) in participants receiving β-alanine + creatine supplementation [14]. Given that creatine is well-documented to support cognitive function, particularly in sleep-deprived or hypoxic conditions, and that β-alanine may mitigate central fatigue contributors, the combined supplementation may offer benefits for maintaining cognitive performance under conditions of physical and mental stress. This finding has potential practical applications for military personnel, endurance athletes, and individuals performing high-cognitive-demand tasks under fatigue, warranting further investigation into the neuromuscular and cognitive effects of co-supplementation.

Overall, the current RCT evidence does not strongly support a meaningful synergistic effect of combined creatine and β-alanine on traditional endurance performance. While β-alanine has demonstrated efficacy in delaying fatigue by improving work at the fatigue threshold and extending TTE, co-supplementation with creatine has not consistently yielded additive benefits in VO_2_max, lactate threshold, or exercise duration to fatigue beyond what β-alanine or endurance training alone can achieve. Creatine supplementation, on its own, has minimal direct impact on classical endurance metrics and does not appear to synergize with β-alanine for these outcomes. There is some evidence that combined supplementation may slightly improve high-intensity endurance thresholds and capacity, but such effects have been inconsistent across studies and generally not statistically superior to single-supplement interventions.

From a practical standpoint, endurance athletes are unlikely to derive substantial additional benefits from creatine and β-alanine co-supplementation beyond those obtained through structured endurance training alone. However, individuals engaged in mixed metabolic events (e.g., HIIT, combat sports, middle-distance races, or team sports requiring repeated sprint efforts) may experience minor improvements in high-intensity endurance capacity with β-alanine alone, although the contribution of creatine in this context remains limited. Future research should explore the effects of co-supplementation in sport-specific endurance applications, particularly in hybrid aerobic-anaerobic sports where sustained high-intensity efforts play a critical role in performance outcomes.

### 4.3. Body Composition

Creatine supplementation is well established for its role in increasing LBM, primarily through enhanced intracellular water retention, improved strength, and greater progressive overload during resistance training [4,33]. In contrast, β-alanine’s potential effects on body composition are more indirect, hypothesized to occur through increased training volume and higher-intensity efforts, leading to greater hypertrophic adaptation. The proposed synergy between these two supplements is that creatine enhances maximal strength, while β-alanine delays fatigue, allowing for a higher training volume—thereby providing both a heavier and a slightly prolonged training stimulus.

Limited evidence from this review suggests that co-supplementation with creatine and β-alanine may potentially improve body composition adaptations; this effect was observed only in Hoffman et al.’s 10-week study, where the creatine + β-alanine group gained approximately 1 kg more lean mass than the creatine-only group and also experienced greater reductions in body fat [12]. These results suggest a potential recomposition effect, where greater muscle mass is accrued without additional fat gain. For athletes, this could imply an improved lean mass-to-fat ratio over a training cycle, thereby optimizing performance and body composition simultaneously.

The underlying mechanism by which β-alanine might contribute to these improvements remains speculative. One hypothesis is that β-alanine’s buffering effect on muscle acidosis allows athletes to sustain higher training volumes or intensities over time, resulting in greater hypertrophic stimulus and potentially increased caloric expenditure [34]. If an athlete can perform additional repetitions before muscular failure or execute high-intensity intervals at a slightly higher power output, these incremental training improvements may accumulate over several weeks, contributing to enhanced lean mass gains and fat loss. Supporting this theory, previous studies have reported that β-alanine supplementation improved training volume in resistance exercise and promoted lean mass gains when combined with HIIT.

Notably, Hoffman et al. observed higher weekly training volume in the squat exercise in the creatine + β-alanine group, potentially explaining the higher lean mass gains in this cohort. However, it is important to acknowledge that not all studies have observed this effect, as some research has failed to find a significant impact of β-alanine on training volume [35,36]. Furthermore, Hoffman et al.’s study design did not include a β-alanine–only group, making it difficult to determine whether the observed benefits were due to a true synergistic effect or simply a larger effect of creatine alone.

Not all studies have demonstrated a significant benefit of co-supplementation with creatine and β-alanine on body composition. A particularly notable finding is the null results reported by Kresta et al., who found that the combination of creatine and β-alanine was not superior to either supplement alone in terms of lean mass or fat mass changes [9]. Several factors may explain these findings, including shorter study duration, lower relative dosage per body weight, and potential sex-specific differences in physiological responses.

Kresta et al. noted that females generally experience smaller increases in lean mass following creatine supplementation compared to males, a trend that was evident in their study, where the creatine-only group exhibited smaller lean mass gains than those typically observed in male-based studies. When juxtaposed with Hoffman et al.’s findings in male participants, it appears plausible that creatine primarily supports muscle energetics, while β-alanine may facilitate greater training volume and quality, potentially maximizing hypertrophic adaptations over time. However, given that Hoffman et al. is the only study to report a significant synergistic effect of co-supplementation on lean mass, these findings should be interpreted with caution.

Some studies suggest that women do not gain as much muscle mass or retain as much water from creatine supplementation as men, potentially due to lower absolute muscle mass, greater baseline intramuscular creatine stores, or hormonal differences. For example, a meta-analysis by Pashayee-Khamene et al. reported that men gained more than twice the amount of lean mass compared to women following creatine supplementation (1.2 kg vs. 0.54 kg) [5].

Additionally, study duration may have influenced the observed outcomes. While Hoffman et al. conducted a 10-week trial, which allowed for more cumulative training adaptations, Kresta et al.’s study lasted only 4 weeks, which may have been insufficient to detect meaningful differences in body composition.

Another important factor is statistical power. Hoffman et al. included 33 participants, and even with this sample size, the between-group differences were only barely significant (*p* < 0.05). In contrast, Kresta et al. divided 32 participants across four groups (~8 per group), significantly reducing statistical power. As a result, the study may have been underpowered to detect modest but potentially meaningful effects of supplementation.

A meta-analysis of 20 RCTs by Ashtary-Larky et al. concluded that, regardless of exercise program, supplementation dose, or study duration, β-alanine supplementation does not significantly increase lean mass or reduce fat mass [37]. This suggests that β-alanine does not have a direct impact on muscle hypertrophy and that the hypertrophic effects observed in creatine + β-alanine co-supplementation studies are primarily attributable to creatine.

Thus, existing evidence does not strongly support the hypothesis that β-alanine enhances the body composition benefits of creatine supplementation. The observed improvements in lean mass in some studies may simply reflect creatine’s well-established role in improving muscle energetics and increasing training capacity, rather than a true synergistic effect with β-alanine.

The current body of evidence regarding the benefits of co-supplementation with creatine and β-alanine for body composition remains limited and inconclusive. Given that previous research has not demonstrated a significant effect of β-alanine on lean mass or fat loss, it is unlikely that β-alanine enhances the body composition benefits of creatine supplementation.

Future studies should focus on longer-duration trials with larger sample sizes to determine whether β-alanine contributes to hypertrophic adaptations when combined with creatine. Additionally, research should explore sex-specific responses and variations in training protocols, as current findings suggest that the potential benefits of co-supplementation may be population-dependent.

In practical terms, athletes seeking to optimize muscle hypertrophy and body composition should prioritize creatine supplementation, as its effects on lean mass and strength are well established. While β-alanine may provide ergogenic benefits for sustained high-intensity efforts, its direct role in body composition improvements remains uncertain. Future research is necessary to clarify whether co-supplementation provides advantages over creatine alone, particularly in specific athletic populations or training contexts.

### 4.4. Dosing Strategies and Interindividual Variability

An important consideration in interpreting the findings of this review is the variation in dosage, duration, and participant characteristics across included studies. For creatine, supplementation protocols included a loading phase (~20 g/day or 0.3 g/kg/day, divided into multiple doses over 5–7 days) followed by a maintenance dose (~5–10 g/day), or in some cases, either a loading-only or maintenance-only protocol. For β-alanine, daily dosages ranged from 3.2 to 6.4 g. Supplementation durations varied between 4 and 10 weeks. Additionally, heterogeneity in participant characteristics—including training status (e.g., trained, recreational, and untrained individuals), sex, and baseline nutritional status—may have influenced the observed outcomes. These variations limit the ability to identify optimal protocols for co-supplementation, highlighting the need for future studies to clarify “how much, how often, and in whom” these supplements are most effective.

Furthermore, individual variability in supplement response—often categorized as “responders” and “non-responders”—is another important factor that may have affected the outcomes assessed in this review. Factors such as genetics [38], muscle fiber composition [39], and baseline dietary creatine [39] or carnosine levels [40] may influence the magnitude of response [39]. For example, Syrotuik and Bell explored responders and non-responders to creatine supplementation [39]. They found that baseline muscle cross-sectional area, a greater proportion of type II muscle fibers, and lower intramuscular creatine stores had greater performance improvements following creatine supplementation and were described as responders. Similarly, beta-alanine responders are more likely to have lower baseline muscle carnosine levels, a greater proportion of Type II muscle fibers, be less well-trained, and follow a vegetarian diet [40]. Importantly, the included RCTs in our systematic review did not stratify or report data based on these variables, which may contribute to the inconsistency and heterogeneity observed across individual studies. Future trials should address this gap by including analyses stratified by response status to better inform individualized recommendations.

### 4.5. Limitations

This systematic review has several limitations that should be acknowledged. First, the total number of included studies (*n* = 7) is relatively small, limiting the ability to draw definitive conclusions regarding the effects of creatine and β-alanine co-supplementation. Additionally, for some key performance and body composition outcomes, only a few studies were available, reducing the reliability and generalizability of the findings. The limited number of studies also precludes the possibility of conducting a meta-analysis, which would have provided a more robust quantitative assessment of the combined effects of these supplements.

Second, another main limitation of this review is the considerable methodological heterogeneity among the included studies. Participant populations varied in terms of training status, sex, and baseline fitness level, and dosing protocols for creatine and β-alanine supplementation were not standardized. These differences limit the ability to draw definitive conclusions and hinder the applicability of findings across all populations.

Third, the duration of most included studies was relatively short, with intervention periods typically ranging from 4 to 5 weeks, except for one longer trial lasting 10 weeks. Given that muscle hypertrophy, body recomposition, and chronic adaptations to training require extended periods, the short intervention durations may have limited the ability to detect meaningful long-term effects of creatine and β-alanine co-supplementation. Future research should focus on longer-duration trials to determine whether the potential synergistic effects of these supplements become more pronounced over time.

Despite these limitations, this review provides valuable insights into the effects of creatine and β-alanine co-supplementation on strength, power, endurance, and body composition. However, future studies with larger sample sizes, longer intervention periods, and more standardized outcome measures are needed to further clarify the efficacy and applicability of co-supplementation in different athletic populations.

## 5. Conclusions

In summary, co-supplementation with creatine and β-alanine may be an effective strategy for enhancing strength, power, and body composition compared to a placebo. However, when comparing creatine and β-alanine combined to either alone, the most consistent benefits tend to enhance high-intensity exercise performance, particularly in activities requiring repeated maximal efforts or sustained anaerobic power output. Although some evidence suggests potential benefits for maximal strength and body composition, the current data remain inconsistent and inconclusive, limiting the ability to draw definitive conclusions regarding a synergistic effect between these supplements. Based on the existing body of research, athletes may potentially benefit from this combination when their primary goal is to enhance high-intensity performance, such as in sprint-based sports, resistance training, or repeated explosive efforts. Conversely, no consistent improvements in aerobic endurance capacity have been observed with creatine and β-alanine co-supplementation, reinforcing the currently limited role of these supplements in oxidative metabolism. Future long-term, well-controlled studies are necessary to further clarify the effects of co-supplementation on different athletic populations and training modalities, ensuring that recommendations are based on reliable and robust evidence. Further, there are large individual responses to creatine and β-alanine supplementation, which may be associated with genetic or baseline differences such as diet, training status, and muscle fiber composition that warrant future considerations.

## Figures and Tables

**Figure 1 nutrients-17-02074-f001:**
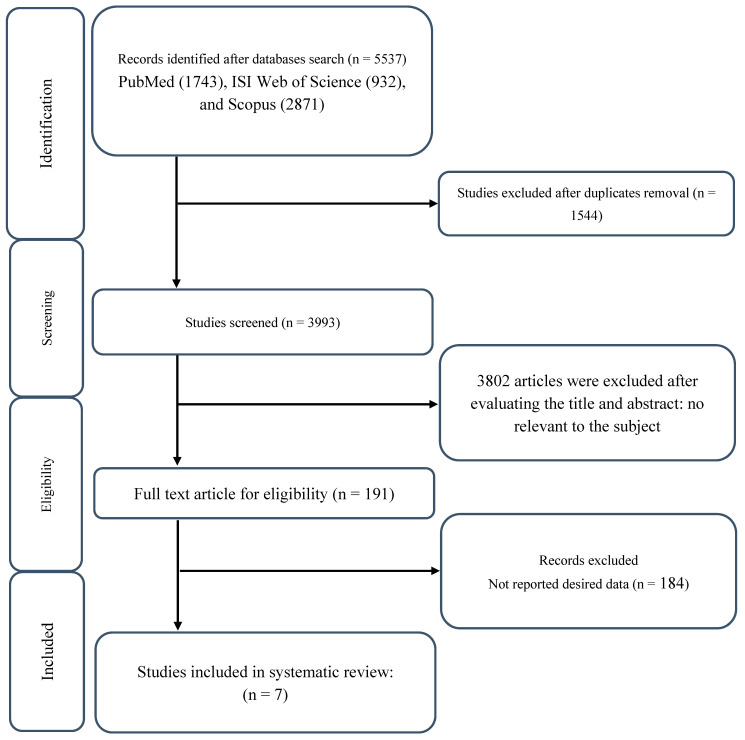
Flow chart of study selection for inclusion trials in the systematic review.

**Figure 2 nutrients-17-02074-f002:**
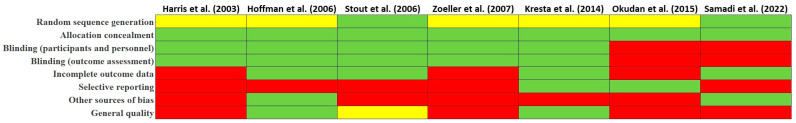
Quality assessment of included studies. Studies with more than two high-risk domains were classified as high risk of bias (red), while those with exactly two high-risk factors were categorized under unclear risk (yellow). Studies exhibiting less than two high-risk domains were considered low risk of bias (green) [9,11,12,14,15,16,17].

**Table 1 nutrients-17-02074-t001:** Characteristics of included studies in meta-analysis.

Author (Year)	Country	Study Design	Participants	Intervention & Sample Size	Duration (Week)	Mean Age	Cr + βA Dosage & Administration	Training Type	Tests	Placebo	Impact of Cr + BA Compared to Cr or BA Alone
Harris et al. (2003) [17]	USA	P, R, PC, DB	28 Healthy men	βA + Cr: 10 Plac + Cr: 9 Plac + Plac: 9	5	24–30 yrs	βA: 4 × 800 mg/d for 5 wks Cr: 4 × 5 g/d in 5th wk	None	Isometric endurance at 50% MVC, and PO4max test	ND	Body composition: ND Anaerobic performance: ↑ Aerobic performance: ND
Hoffman et al. (2006) [12]	USA	P, R, PC, DB	33 Male strength power athletes	βA + Cr: 11 Cr: 11 Plac: 11	10	NR	Cr: 10.5 g/day; βA: 3.2 g/day	RT	BC (DEXA), 1 RM for squat and bench press, Wingate anaerobic power test and 20 jump power test	Dextrose (10.5 g/day)	Body composition: ↑ Anaerobic performance: ↑ Aerobic performance: ND
Stout et al. (2006) [15]	USA	P, R, PC, DB	51 Untrained healthy men	βA + Cr: 14 βA: 12 Cr: 12 Plac: 13	4	24.5	Day 1–6: βA: 4 × 1.6 g/d Cr: 4 × 5.25 g/d Day 7–28: βA: 2 × 1.6 g/d Cr: 2 × 5.25 g/d	None	Cycle ergometry test, PWC_FT_	Dextrose (34 g/dose)	Body composition: ND Anaerobic performance: ND Aerobic performance: ↔
Zoeller et al. (2007) [11]	USA	P, R, PC, DB	55 Untrained healthy men	βA + Cr: 16 βA: 14 Cr: 12 Plac: 13	4	24.5	Day 1–6: βA: 4 × 1.6 g/d Cr: 4 × 5.25 g/d Day 7–28: βA: 2 × 1.6 g/d Cr: 2 × 5.25 g/d	None	Graded exercise test to determine VO_2_peak, VT and LT	Dextrose (34 g/dose)	Body composition: ND Anaerobic performance: ND Aerobic performance: ↔
Kresta et al. (2014) [9]	USA	P, R, PC, DB	32 Recreationally active females	βA + Cr: 9 βA: 8 Cr: 8 Plac: 7	4	21.5	βA: 0.1 g/kg/day Cr: 0.3 g/kg/day (1 week), then 0.1 g/kg/day	None	BC (DEXA), Wingate tests to determine PP, MP & FI, Graded exercise test to determine VO_2_peak, VT and LT	Maltodextrin (0.1 g/kg/day for 4 week) + dextrose (0.3 g/kg/day for 1 week)	Body composition: ↔ Anaerobic performance: ↔ Aerobic performance: ↔
Okudan et al. (2015) [16]	Turkey	P, R, PC	44 Untrained healthy men	βA + Cr: 11 βA: 11 Cr: 11 Plac: 11	4	20–22	Day 1–22: βA: 2 × 1.6 g/d Cr: 2 × 5 g/d Day 23–28: βA: 4 × 1.6 g/d Cr: 4 × 5 g/d	None	Wingate Test to determine PP, MP & FI, Graded exercise test to determine VO_2_peak, VT and LT	Maltodextrin (20 g/day for day 1–22 and 40 g/day for day 23–28)	Body composition: ND Anaerobic performance: ↑ Aerobic performance: ND
Samadi et al. (2022) [14]	Iran	P, R, PC	20 Active male military personnel	βA + Cr: 10 βA: 10	4	21.5	βA: 6.4 g/day for 28 days; Cr: 0.3 g/kg/day for 7-day loading phase	Military training	RAST, 1RM for bench press and leg press, Vertical Jump Test, and SCET	Equal grams of rice flour	Body composition: ND Anaerobic performance: ↑ Aerobic performance: ND

P = parallel; R = randomized; PC = placebo control; DB = double blind; Cr = creatine; βA = β-alanine; Plac = placebo; MVC = Maximum Voluntary Contraction; PO4max = power output during a 4 min all-out maximal ergometer exercise; ND = not determined; RT = resistance training, BC = body composition, DEXA = dual energy X-ray absorptiometry; 1RM = 1-repetition maximum; PWC_FT_ = physical working capacity at neuromuscular fatigue threshold; VT = ventilatory threshold; LT = lactate threshold; PP = peak power; MP = mean power; FI = fatigue index; RAST = running anaerobic sprint test; SCET = Simulated Casualty Evacuation Test; ↑ = improvement; ↔ = no significant change.

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
