# Peer review of "Effects of Creatine and β-Alanine Co-Supplementation on Exercise Performance and Body Composition: A Systematic Review"

_nutrients, 2025, doi:10.3390/nu17132074_

Round 1

Reviewer 1 Report

Comments and Suggestions for Authors

Major comments:

  1. Include an explicit hypothesis or research question.
  2. Briefly justify the decision to conduct a qualitative synthesis instead of a meta-analysis.
  3. Clarify the criteria for exclusion of multi-ingredient studies (what if those ingredients were matched between groups?).
  4. Add a summary table of main findings with effect direction and quality of evidence.
  5. Focus more on discussing the included studies, not the excluded ones.
  6. Reorganize the discussion around the main review objectives (anaerobic performance, body composition, aerobic performance).

Minor comments

  1. In conclusion: Use more cautious language, stressing that the additional benefits of co-supplementation are modest and context-dependent (type of exercise, population).
  2. Reference formatting should be standardized. Some studies are incorrectly or inconsistently cited (e.g., [3], [5]).

Author Response

Reviewer 1.

The manuscript presents a well-structured systematic review with clear objectives on the co-supplementation of creatine and β-alanine. The review addresses a relevant topic in the field of sports nutrition. However, some methodological weaknesses, writing inconsistencies, and analytical gaps should be addressed or clarified before the manuscript can be considered for publication.

Major comments:

  1. Include an explicit hypothesis or research question.

Authors: Thank you for your suggestion. We have now added an explicit research question and hypothesis at the end of the Introduction section to clarify the objective of our systematic review and meta-analysis.

  1. Briefly justify the decision to conduct a qualitative synthesis instead of a meta-analysis.

Authors: Thank you for your comment. Due to the small number of studies available for each specific outcome variable and the high heterogeneity in performance outcomes and testing protocols, we conducted a qualitative synthesis rather than a meta-analysis. We have added a clarifying sentence to the Methods section to explain this decision.

  1. Clarify the criteria for exclusion of multi-ingredient studies (what if those ingredients were matched between groups?).

Authors: Thank you for your comment. We excluded multi-ingredient studies when the comparison between groups did not allow the isolation of the effect of creatine and β-alanine. However, if both groups received the same co-intervention and the only difference was the presence or absence of creatine and β-alanine, such studies were eligible for inclusion. We have added a clarifying sentence to the Methods section to explain this decision.

  1. Add a summary table of main findings with effect direction and quality of evidence.

Authors: Thank you for your valuable suggestion. As advised, we have added a summary of the main findings, including the direction of the effect (positive, negative, or no effect), to Table 1 to enhance clarity and accessibility.

However, regarding the quality assessment of evidence using the GRADE framework, we respectfully did not apply GRADE due to the limited number of studies available for most outcomes. Specifically, for several performance and body composition markers, only 1–2 studies were available, making it impossible to conduct a meaningful and reliable GRADE evaluation. As GRADE requires a sufficient number of studies to assess consistency, precision, and other domains, performing it under such conditions could be misleading.

We hope this explanation is acceptable and appreciate your understanding.

  1. Focus more on discussing the included studies, not the excluded ones.

Authors: Thank you for your valuable comment. We have revised the Discussion section to ensure that the main focus remains on the included studies. Any prior emphasis on excluded studies has been minimized. The only excluded study mentioned (Li et al.) was briefly discussed due to the limited number of included studies and its relevance to the topic. However, the interpretation throughout the revised section now centers more clearly on the findings and quality of the included evidence.

Reorganize the discussion around the main review objectives (anaerobic performance, body composition, aerobic performance).

Authors: Thank you for your helpful suggestion. We have revised the Discussion section to structure it clearly around the main review objectives: anaerobic performance, body composition, and aerobic performance. This reorganization improves clarity and enhances the relevance of our interpretations.

Minor comments

  1. In conclusion: Use more cautious language, stressing that the additional benefits of co-supplementation are modest and context-dependent (type of exercise, population).

Authors: Thank you for your valuable feedback. We have revised the conclusion to adopt a more cautious tone, emphasizing that the additional benefits of creatine and β-alanine co-supplementation appear to be modest and may vary depending on the exercise type and participant characteristics.

  1. Reference formatting should be standardized. Some studies are incorrectly or inconsistently cited (e.g., [3], [5]).

Authors: Thank you for pointing this out. All references have been carefully reviewed and reformatted to ensure consistency with the journal’s referencing style.

Reviewer 2 Report

Comments and Suggestions for Authors

I read the article. I think this is an important subject and a systematic review give to all the doctors more informations. The author show us the most relevant RCT, the results and make a complete analysis of all the data. 

Author Response

Reviewer 2.

I read the article. I think this is an important subject and a systematic review give to all the doctors more informations. The author show us the most relevant RCT, the results and make a complete analysis of all the data.

Authors: Thank you very much for your positive and encouraging feedback. We truly appreciate your recognition of the importance of this topic and the value of our systematic analysis. Your kind words motivate us to continue improving the quality of our research.

Reviewer 3 Report

Comments and Suggestions for Authors

Peer Review Report

Title:

The present study investigates the effects of creatine and β-alanine supplementation on exercise performance and body composition. A Systematic Review

A number of methodological, interpretative and ethical concerns have been identified that limit the robustness and scientific neutrality of the conclusions.

The authors do not exaggerate the benefits, and they clearly acknowledge the absence of synergistic effects in certain domains (e.g., aerobic capacity), BUT

the following issues are of major concern:

  1. Conflict of Interest and Scientific Neutrality

It has been noted that several authors have substantial commercial affiliations with companies that specialize in the marketing of creatine and β-alanine supplements. While these conflicts are disclosed, they are not critically addressed in terms of how they may bias study selection, interpretation, or framing of results. The tone of the paper occasionally approaches a state of promotion.

It is recommended that… It is recommended that an independent statistical and content review be conducted to ensure neutrality or enhance disclosure mitigation.

  1. The present study is characterized by an absence of meta-analysis.

Despite the heterogeneity cited, the data could support subgroup or sensitivity analyses. The decision to eschew quantitative synthesis has the effect of weakening the conclusions.

It is recommended that… It is imperative that a partial meta-analysis is incorporated, or, alternatively, that at least forest plots are included for key outcomes.

  1. The overinterpretation of evidence of a limited nature

The article employs language of ambiguity, such as 'may' and 'potentially', despite the absence of statistically significant results. This undermines the scientific rigor of the study.

It is recommended that… It is recommended that the use of scientific language be elevated to a more conservative and precise standard.

  1. The present study explores the presence of methodological variability within the included studies.

Participant populations, durations, and dosing protocols vary extensively. There is an absence of a clearly defined stratified analysis that can adequately address this heterogeneity.

It is recommended that… The incorporation of stratified tables and a more thorough discussion of subpopulations is recommended.

  1. The present study is lacking in originality and innovation.

The findings of the study largely serve to reaffirm existing knowledge in the field, offering no novel mechanistic insights or evidence that has the potential to effect meaningful change in clinical practice.

It is recommended that… The information should be presented as an update and the limits of contribution clarified.

It is important to note the following minor points:

Errors were observed in the author’s affiliations and contact information.

It is evident that while the figures have been referenced, they remain invisible within the PDF.

It is important to note that Google Scholar should not be cited as a formal scientific database.

The final recommendation is as follows:

A substantial revision is imperative.

The manuscript requires a more cautious interpretation of findings, stronger justification of methodology, and a neutral tone given disclosed conflict of interest. These revisions have the potential to serve as a concise update on the current evidence for creatine and β-alanine co-supplementation.

Author Response

Reviewer 3.

Title:

The present study investigates the effects of creatine and β-alanine supplementation on exercise performance and body composition. A Systematic Review

A number of methodological, interpretative and ethical concerns have been identified that limit the robustness and scientific neutrality of the conclusions.

The authors do not exaggerate the benefits, and they clearly acknowledge the absence of synergistic effects in certain domains (e.g., aerobic capacity), BUT the following issues are of major concern:

  1. Conflict of Interest and Scientific Neutrality

It has been noted that several authors have substantial commercial affiliations with companies that specialize in the marketing of creatine and β-alanine supplements. While these conflicts are disclosed, they are not critically addressed in terms of how they may bias study selection, interpretation, or framing of results. The tone of the paper occasionally approaches a state of promotion.

It is recommended that… It is recommended that an independent statistical and content review be conducted to ensure neutrality or enhance disclosure mitigation.

Authors: We appreciate the reviewer’s comment. However, we would like to emphasize that this manuscript is a systematic review of previously published studies. We did not intervene in or manipulate any original data. All data synthesis was based strictly on available evidence from independent trials conducted by other researchers.

While some of the authors have disclosed commercial affiliations, these affiliations had no influence on the selection, interpretation, or reporting of results. The entire process—from literature search to data extraction and analysis—was conducted following rigorous scientific methodology and transparency, ensuring objectivity and neutrality in presenting the current body of evidence.

Nevertheless, we have carefully reviewed the manuscript again to ensure that the language remains neutral and non-promotional throughout.

  1. The present study is characterized by an absence of meta-analysis.

Despite the heterogeneity cited, the data could support subgroup or sensitivity analyses. The decision to eschew quantitative synthesis has the effect of weakening the conclusions.

It is recommended that… It is imperative that a partial meta-analysis is incorporated, or, alternatively, that at least forest plots are included for key outcomes.

Authors: We acknowledge the reviewer’s suggestion and fully understand the value of a quantitative synthesis. However, the primary limitation in conducting a meta-analysis or even subgroup/sensitivity analyses was the extremely limited number of studies per outcome marker.

For some of the outcomes assessed (e.g., specific measures of body composition, anaerobic and aerobic performance), only one or two studies were available, and the variations in study design, supplementation protocols, and outcome definitions further exacerbated heterogeneity. Given these constraints, performing a meaningful or statistically valid meta-analysis was not feasible without risking misleading conclusions.

Therefore, we opted for a qualitative synthesis to maintain scientific rigor and avoid overstating the strength of evidence. We hope this justification clarifies our decision to refrain from quantitative pooling in the current version of the manuscript.

  1. The overinterpretation of evidence of a limited nature

The article employs language of ambiguity, such as 'may' and 'potentially', despite the absence of statistically significant results. This undermines the scientific rigor of the study.

It is recommended that… It is recommended that the use of scientific language be elevated to a more conservative and precise standard.

Authors: Thank you for your thoughtful comment. We acknowledge the importance of maintaining scientific rigor and avoiding overinterpretation. We have carefully reviewed the manuscript and revised sentences where language may have implied stronger evidence than currently exists. Specifically, we have adjusted statements that previously used terms such as “may” or “potentially” when the findings were not statistically significant or were based on limited evidence. We now emphasize the inconsistency and limitations of the existing data more clearly to ensure a conservative and precise presentation of the results.

  1. The present study explores the presence of methodological variability within the included studies.

Participant populations, durations, and dosing protocols vary extensively. There is an absence of a clearly defined stratified analysis that can adequately address this heterogeneity.

It is recommended that… The incorporation of stratified tables and a more thorough discussion of subpopulations is recommended.

Authors: We fully agree with the reviewer that there is considerable methodological heterogeneity among the included studies, including differences in participant characteristics, intervention durations, and dosing protocols. This is indeed one of the key limitations of our review. We will acknowledge this point explicitly in the limitations section of the manuscript and consider stratified interpretations where feasible in our discussion.

  1. The present study is lacking in originality and innovation.

The findings of the study largely serve to reaffirm existing knowledge in the field, offering no novel mechanistic insights or evidence that has the potential to effect meaningful change in clinical practice.

It is recommended that… The information should be presented as an update and the limits of contribution clarified.

Authors: We acknowledge that our study does not aim to introduce novel mechanistic pathways or clinical innovations. Instead, the primary objective was to provide a concise and updated synthesis of the current literature on creatine and β-alanine co-supplementation, particularly focusing on athletic performance and body composition outcomes.

It is important to note the following minor points:

Errors were observed in the author’s affiliations and contact information.

Authors: The authors' affiliations and contact information have been carefully reviewed and corrected.

It is evident that while the figures have been referenced, they remain invisible within the PDF.

Authors: In the updated version of the manuscript, all referenced figures have been properly inserted and are now fully visible in the revised file.

It is important to note that Google Scholar should not be cited as a formal scientific database.

Authors: We agree that Google Scholar is not a formal scientific database and have removed the reference to it accordingly.

The final recommendation is as follows:

A substantial revision is imperative.

Authors: We appreciate the reviewer’s thorough evaluation. A substantial revision has been made to address all comments and enhance the clarity, neutrality, and scientific rigor of the manuscript.

The manuscript requires a more cautious interpretation of findings, stronger justification of methodology, and a neutral tone given disclosed conflict of interest. These revisions have the potential to serve as a concise update on the current evidence for creatine and β-alanine co-supplementation.

Authors: We appreciate the reviewer’s constructive feedback. In response, we have revised the manuscript to adopt a more cautious and balanced interpretation of the findings, strengthened the justification of our methodology, and ensured a more neutral tone throughout the text. We hope these revisions improve the clarity and objectivity of the manuscript and allow it to serve as a concise and informative update on the current evidence regarding creatine and β-alanine co-supplementation.

Round 2

Reviewer 1 Report

Comments and Suggestions for Authors

Accept in present form

Author Response

Thank you very much for your kind acceptance of our manuscript.